# An Adaptive Defense against Adversarial Patch Attacks for Vision Transformers

## Abstract

Vision Transformers (ViTs) have become the prominent architecture for various computer vision tasks due to their superior ability to capture long-range dependencies through the self-attention mechanism. However, recent research indicates that ViTs are highly susceptible to carefully crafted adversarial patch attacks, presenting a significant challenge for practical deployment, particularly in security-critical applications. Existing approaches towards robust ViT frameworks often sacrifice clean accuracy and/or achieve suboptimal robustness, likely due to their uniform handling of diverse input samples. In this paper, we present *NeighborViT*, a novel adaptive defense framework specifically designed to counter adversarial patch attacks for ViTs. NeighborViT stands out by detecting and categorizing different types of attacks on inputs and applying adaptive, tailored defense mechanisms for each type of attack. To realize effective attack detection, categorization, and mitigation, NeighborViT explores the information in neighbor patches of the target patch and strategically employs them for defense. Our experimental results on the ImageNet dataset using various state-of-the-art ViT models demonstrate that NeighborViT significantly enhances robust accuracy without compromising clean accuracy. Our code is available at `https://anonymous.4open.science/r/NeighborViT-8255`.

## 1 Introduction

Vision Transformers (ViTs) (Dosovitskiy et al., 2021) have become the leading architecture in various computer vision tasks, such as image classification (Zhu et al., 2023), segmentation (Ye et al., 2019), and generation (Chen et al.). However, they recently show heightened vulnerability to adversarial patch attacks (Brown et al., 2017; Gu et al., 2022; Fu et al., 2022; Lovisotto et al., 2022; Yuan et al., 2024). These attacks, which introduce small but strategically placed patches to an image, exploit ViTs' attention mechanism, leading to significant model misclassifications. For example, only 0.5% modifications to the input image can degrade the model's performance to 0% (Lovisotto et al., 2022). This vulnerability underscores a fundamental weakness in the current design of ViTs and raises concerns regarding their reliability in real-world applications.

Various works have been proposed to enhance the robustness of ViTs. One line of work treats the model as a black box and analyzes the model inputs/outputs to mitigate adversarial patch attacks (Xiang et al., 2022; Tarchoun et al., 2023; Yang et al., 2024). In contrast, another line of work leverages the unique self-attention mechanism in ViTs to limit the impact of abnormal attention of adversarial patches (Yu et al., 2023; Liu et al., 2023; Mu & Wagner, 2021). Despite their effectiveness, many of the above studies (Yu et al., 2023; Liu et al., 2023; Kim et al., 2023) process clean and malicious inputs indistinguishably, which inevitably harms the model's clean accuracy. Although some studies (Xiang et al., 2022; Tarchoun et al., 2023; Yang et al., 2024) can discern adversarial inputs, they rely on computationally expensive detectors to manage the challenges posed by unknown attack sizes and positions, and they treat all adversarial inputs equally without considering the impact of different attacked locations, resulting in limited improvements in robustness.

In this paper, we highlight the importance of distinguishing different types of inputs and introduce *NeighborViT*, an input-adaptive defense framework for ViTs against adversarial patch attacks. NeighborViT not only detects adversarial inputs from clean ones but also categorizes different at-

tack types, subsequently adopting tailored defense strategies. The key insight behind NeighborViT is leveraging information from neighboring patches of target patches for effective attack detection, categorization, and mitigation. Specifically, adversarial patches typically exhibit high pixel discontinuity compared to clean patches. By analyzing pixel-level discontinuity differences between target and neighboring patches, we develop a lightweight and accurate algorithm to detect and locate attacks. Such detection enables us to maintain the target models' clean accuracy. To improve robustness, we also distinguish between *catastrophic* and *non-catastrophic attacks* based on whether they occur in essential or non-essential areas for classification. The categorization is achieved by replacing the adversarial patch with its neighbors and observing model output variations. Non-catastrophic attacks show more consistent outputs since they do not harm the essential features for classification.

After categorization, we develop specialized defenses for different types of attacks. For non-catastrophic attacks, we replace adversarial patches with neighboring patches, which entirely remove adversarial information and preserve the essential feature for classification. Such replacement cannot be applied to catastrophic attacks, as this would result in the loss of essential features. Hence, we design a fine-grained attention suppression mechanism instead to suppress the adversarial attention.

Our contributions are summarized as follows:

- We develop NeighborViT, a novel robust ViT framework that protects ViT against adversarial patch attacks. We utilize neighbor information to categorize model inputs and design tailored defenses for each category. This adaptive defense strategy enables us to achieve high robustness while maintaining clean accuracy.
- We explore the pixel-level discontinuity differences between adversarial patches and the neighboring patches and present a model-agnostic attack detector. Our detector can accurately and efficiently detect and localize adversarial patches of unknown sizes.
- We show the necessity to differentiate between catastrophic and non-catastrophic attacks and propose an essential/non-essential area detector for this. The detection is enabled by exploiting model prediction variations when adversarial patches are replaced with different neighboring patches.
- We propose a fine-grained attention suppression algorithm for catastrophic attacks and an adversarial patch reconstruction method for non-catastrophic attacks. These tailored defenses enable optimized robust accuracies.

To evaluate our method, we conduct extensive experiments on 12 representative ViT models across various state-of-the-art attack approaches. Our results show that we achieve the best robust performance while maintaining clean accuracy compared to other methods.

## 2 BACKGROUND & RELATED WORKS

**Vision Transformer**: The Vision Transformer (ViT) (Dosovitskiy et al., 2021), inspired by NLP models like BERT (Devlin et al., 2019), introduces self-attention to image classification, offering an alternative to traditional CNNs. ViTs excel in tasks requiring broader context understanding, as they avoid CNNs' reliance on local receptive fields. Notable ViT models include ViT (Dosovitskiy et al., 2021), DeiT (Touvron et al., 2021), BiFormer (BiF) (Zhu et al., 2023), and TransNeXt (TNX) (Shi, 2024), with TNX and BiF outperforming CNNs by over 15% in classification tasks. A typical workflow of ViTs is as follows. ViTs split input images into patches, transforming them into embeddings. These embeddings, along with positional encodings, are fed into a transformer encoder composed of multiple transformer blocks. In each block, embeddings first pass through the Multi-Head Self Attention (MHSA) layer, where they are converted into queries ($Q$), keys ($K$), and values ($V$). Subsequently, the attention output for each head is calculated as

$$\text{Attention}(Q, K, V) = \text{softmax}\left(\frac{QK^T}{\sqrt{d_k}}\right) V \tag{1}$$

where $d_k$ is the vector dimension. After that, the outputs from all attention heads are concatenated and linearly transformed, producing the output of the MHSA layer. The output is then passed through residual connections and layer normalization before being fed into the MLP layer to incorporate nonlinear information. Through layer-by-layer connections, the final classification is

achieved through an MLP head using the representation of a unique CLS token. This structured approach enables ViTs to effectively leverage self-attention for superior classification performance.

**Adversarial patch attacks**: Brown et al. (Brown et al., 2017) first introduce adversarial patch attacks, which limit the attack region to a patch area. In Appendix A.1, we show some examples of patch attacks. Initially targeting CNNs, adversarial patch attacks have now expanded to Vision Transformers (ViTs). Recent studies (Gu et al., 2022; Fu et al., 2022) reveal ViTs' vulnerability to patch attacks, exploiting ViTs' need to partition images into patches for attention computation. One line of ViT's patch attack focuses on designing loss functions that target only the model's output, utilizing gradient to optimize the adversarial patch aligned with the input patches of ViTs, such as Token-attack (Joshi et al., 2021) and ViTRPP (Gu et al., 2022). Based on the global reasoning of attention being the source of the vulnerability of ViT to patch attacks, another line of work not only utilizes the model's output but also incorporates attention-aware loss. Fu et al. propose (Fu et al., 2022) Patch-Fool with integrated attention-aware patch selection technique and attention-aware loss design. Subsequently, Lovisotto et al. (Lovisotto et al., 2022) observe that using post-softmax attention scores as a loss in Patch-Fool leads to the issue of smaller gradients, thus limiting the attack's potential. Therefore, they proposed Attention-fool, which designs the loss using pre-softmax attention scores to avoid this problem.

**Defense methods for ViTs**: Defense strategies against ViT's patch attacks can be divided into model-agnostic and ViT-specific methods. Model-agnostic defenses treat the model as a black box and are generally applicable to both CNNs and ViTs; for instance, PatchCleanser (Xiang et al., 2022) uses two rounds of moving window masking and output analyzing to get the correct answer; Jedi (Tarchoun et al., 2023) identifies adversarial patches using entropy analysis and an autoencoder, exploit the fact that adversarial patches have higher entropy than natural images. To get the correct classification result, Jedi applies a pixel reconstruction method on attacked images. ViT-specific defenses leverage attention mechanisms; for example, Robust Self-Attention Layer (Mu & Wagner, 2021) detects and masks outlier tokens based on their value vector; ARMRO (Liu et al., 2023) detects and masks adversarial patches by identifying the layers where the adversarial token's score becomes most prominent, based on its varying behavior across different layers; RTA (Yu et al., 2023) addresses the issue of adversarial patches attracting excessive attention in ViTs by applying a restriction operation on the attention matrix.

However, most existing methods do not differentiate between clean and adversarial inputs, limiting their robustness and clean accuracy. Even when adversarial patch attacks are identified in some methods, they do not classify the types of attacks (e.g., catastrophic and non-catastrophic attacks), limiting their robustness. In contrast, our approach adaptively processes different kinds of input, resulting in an improved model's clean accuracy and robustness. The necessity of adaptive processing is elaborated further in the following section.

## 3 Design of NeighborViT

In Fig. 1 *Top*, we introduce our adaptive defense framework, named NeighborViT, which distinguishes different types of adversarial patch attacks and adopts corresponding defense methods. The framework comprises an attack detector, an essential/non-essential area detector, image reconstruction for non-catastrophic attacks, and attention suppression for catastrophic attacks. The catastrophic attacks represent the attacks occurred in the essential areas and non-catastrophic attacks represent the attacks located in the non-essential areas. For any input sample, we first conduct attack detection through the lightweight and effective attack detector (**detection**). If the input is clean, it will be directly inputted into the ViT model for classification. Conversely, if the input contains adversarial patches, we use the essential/non-essential area detector to identify whether the attack occurs in the essential area or non-essential area (**categorization**). After that, we design different defense methods to mitigate the impact of adversarial patches (**mitigation**). If the attack is located in the essential area, we use fine-grained attention suppression to defend catastrophic attacks. If it occurs in the non-essential area, we remove the adversarial patches and reconstruct the image. Notably, all the above process, *i.e.*, detection, categorization, and mitigation, are enhanced with the information explored in the neighbors of target patches. With this framework, we can achieve superior robust performance against adversarial patch attacks while maintaining clean accuracy.

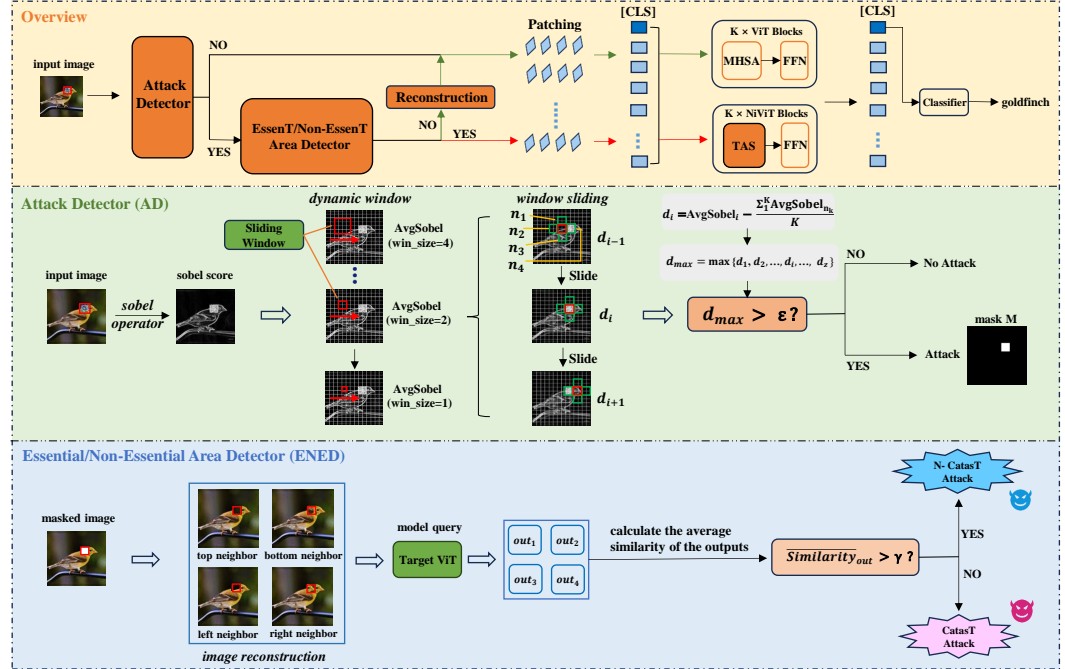

Figure 1: **Overview of NeighborViT**. *Top*: The framework of NeighborViT. Each input sample is categorized into clean samples, catastrophic attacks, and non-catastrophic attacks by the attack detector and essential/non-essential area detector. We do not modify clean inputs and adopt different defense methods (e.g., reconstruction or attention suppression mechanisms TAS) for different attack types. *Middle*: The details of the attack detector, which divides the input samples into clean samples and adversarial patch attacks. *Bottom*: The details of the essential/non-essential area detector, which distinguishes whether the attack occurs in essential areas (i.e., catastrophic attacks) or non-essential areas (i.e., non-catastrophic attacks). We add red boxes on the images for better visualization.

## 3.1 ATTACK DETECTOR

Why do we need to design a novel attack detector? The answer lies in two aspects. On the one hand, in existing research of enhancing the robustness of ViTs, some methods (Yu et al., 2023; Liu et al., 2023; Kim et al., 2023) overlook whether an attack has occurred and treat all the input samples equally, which often compromises the clean accuracy of the target model. On the other hand, although other approaches (Xiang et al., 2022; Yang et al., 2023; Tarchoun et al., 2023) consider to detect adversarial patches, they either need to train an auxiliary model or query the target model multiple times to deal with the challenges posed by unknown adversarial patch size and location, leading to significant costs. In this paper, we present a lightweight attack detector that can accurately detect adversarial patches of unknown locations and sizes.

Our motivation stems from the low pixel continuity inside adversarial patches. The generation process of the adversarial patches neglects the relationships between pixels and results in a noticeable pixel gradient. This gradient can be detected through traditional image processing tools (Sobel et al., 1968; Prewitt et al., 1970; Canny, 1986). In particular, the sobel operator (Sobel et al., 1968), designed for edge detection, utilizes horizontal and vertical kernels to detect pixel variations in both directions and effectively highlights pixels with high gradients. Meanwhile, the pixels inside adversarial patches also have high gradients that can be detected by the operator. We present an example to show the effect of the sobel operator on adversarial patches in Fig. 1 *Middle* (more examples can be seen in Appendix A.2), where the white areas represent higher sobel scores. It indicates that the adversarial patches show the most salient gradient, while the pixel gradients within clean patches are comparatively smaller. This contrast can be used for adversarial patch detection.

To detect and locate adversarial patches, we first use the above sobel operator to get the sobel score of each pixel and calculate average sobel score of pixels in each patch, denoted as $AvgSobel$. The patch with a high average sobel score is likely to be an adversarial patch. However, some clean

patches also exhibit relatively high $AvgSobel$ values, relying exclusively on the $AvgSobel$ score for detection and localization may lead to wrong results. To address this, we introduce a new distance score $d$ that not only considers the $AvgSobel$ of the current patch but also incorporates the $AvgSobel$ of patches inside neighbor patches, denoted as $AvgSobel_{n_k}$, where the $n_k$ represents the neighbor patches. The distances $d$ of all patches are combined. The computation of $d$ is illustrated in Eq. 2 and $K$ represents the number of the sampled neighbor patches.

$$d = AvgSobel - \frac{\sum_{k=1}^{K} AvgSobel_{n_k}}{K} \qquad (2)$$

$$d_{max} = \max_{d_i \in \mathbf{D}} \{d_1, d_2, \cdots d_i, \cdots, d_z\} \qquad (3)$$

Since the location of the adversarial patches is unknown, we propose a sliding window method to scan for suspicious patches across the image. First, we assume that the sliding window size and attack patch size are equal to the model's input patch size. For the $i_{th}$ slide, we sample the neighbors of current window and calculate the above distance $d_i$. We group all $d$ into vector $\mathbf{D}$. Second, when finishing the window sliding process, we get the maximum distance $d_{max}$ through Eq. 3, where $z$ represents the total number of slides. At last, if the maximum distance $d_{max}$ exceeds the threshold $\varepsilon$, we deem that we have detected and located the adversarial patches and can obtain the mask $\mathbf{M}$ of the adversarial patches; otherwise, the input sample is clean.

However, in practice, the size of the adversarial patches is often inconsistent with the input patch size of the model, and their sizes are unknown. To accurately locate the adversarial patches, we introduce a dynamic window size strategy during the window sliding process(Fig. 1 *Middle*). We start with a large window size and conduct one round of sliding for each window size. In this case, we calculate the mean of all pixel gradients within the variable window. If no adversarial patch is detected in the current sliding round, we gradually reduce the window size and continue to the next slide. If no adversarial patch is detected after the last round of sliding (i.e., window size equals to the input patch size of the model), we deem that the input sample is clean. The detailed detection algorithm is provided in Appendix A.4. Notably, our attack detector does not require to train auxiliary models or query the target model, thereby incurring no additional cost.

## 3.2 ESSENTIAL/NON-ESSENTIAL AREA DETECTOR

Current defense studies (Yu et al., 2023; Tarchoun et al., 2023; Liu et al., 2023) often do not distinguish different types of adversarial patch attacks and treat them equally. However, we need to utilize different defense measures to achieve better robust performance. Specifically, if the attack occurs in the non-essential areas (non-catastrophic attacks), which does not contain essential features for model classification, we can completely remove these attacked information to achieve strong robust performance of the model. To demonstrate this, we compare the effect of the two methods (i.e. remove adversarial patches and RTA (Yu et al., 2023)) when attacks are located in the non-essential areas, where RTA uses global attention for attention suppression, which is a typical method that still retains the attack information. Specifically, we select four rep-

Table 1: Distinguishing different types of adversarial patch attacks is important (robust accuracy (%) for attacks located in essential and non-essential areas is reported. *Left*: attacks located in non-essential areas; *Right*: attacks located in essential areas).

| Model | Defense | Attack Methods | | |
|---|---|---|---|---|
| | | ViTRPP | Patch-F | Attention-F |
| DeiT-S | RTA | 52.8/**64.2** | 51.3/**61.6** | 50.4/**59.8** |
| | Removal | **67.3**/53.3 | **63.8**/50.7 | **61.0**/51.5 |
| ViT-S | RTA | 43.4/**56.2** | 41.8/**54.7** | 40.5/**54.2** |
| | Removal | **61.8**/45.3 | **62.6**/42.5 | **60.9**/41.3 |
| BiF-S | RTA | 56.4/**64.3** | 61.5/**65.2** | 56.4/**65.1** |
| | Removal | **72.6**/53.1 | **66.8**/59.8 | **64.3**/56.6 |
| TNX-S | RTA | 69.3/**73.1** | 69.0/**73.9** | 67.4/**69.7** |
| | Removal | **74.6**/67.2 | **73.4**/68.8 | **70.5**/65.2 |

resentative ViT models and 1,500 non-catastrophic attacks generated by Patch-Fool (Fu et al., 2022). We set the attack patch size to twice the model's input patch size. The results in Table 1 *Left* show that removing adversarial patches achieves better robustness, suggesting that removal is a superior defense for non-catastrophic attacks.

Is it appropriate to completely remove adversarial patches if they are located in essential areas? To answer this question, we conduct an experiment to compare the effect of the two methods *i.e.*, remove adversarial patches and RTA, when attacks are located in the essential areas. Specifically, we select 1,500 catastrophic attacks generated by Patch-Fool, and the other experimental settings are the same as those in the non-catastrophic attacks experiment. The results in Table 1 *right* are

different from those of non-catastrophic attacks and show that attention suppression achieves better robustness. To explore the reason, we visualize the attention difference between these two methods in Fig. 2. Current attack methods (Fu et al., 2022; Lovisotto et al., 2022) often amplify the key vectors to achieve better attack performance, and red squares in Fig. 2 *Left* represent the attacked token. The suppression method only suppresses the key vectors of adversarial patches while maintaining **q** and **v** vectors, leading to less impact on the attention calculation process. In contrast, as shown in Fig. 2 *Right*, removing the adversarial token also affects the query and value vectors and has a greater impact on attention calculation (the red squares represent the affected vectors). This phenomenon is also verified in Appendix A.3, which shows the impact of these two methods on the attention map. The

Figure 2: **The effect of attention suppression and removal for attacks occurred in essential areas on the attention mechanism**. $X_1$ denotes the adversarial token and $X_2$, $X_3$ are clean tokens. *Left*: suppress the attention of $X_1$; *Right*: remove token $X_1$.

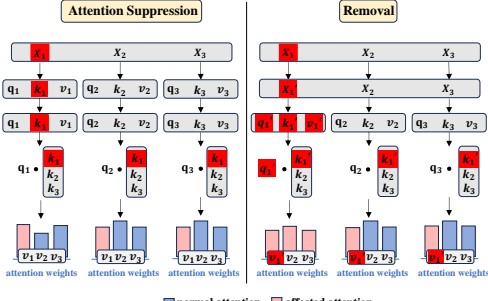

above experimental results and analysis suggest that adversarial patches have useful information for classification and cannot be completely removed for catastrophic attacks.

Based on the above experimental results and analysis, we need to distinguish essential and non-essential areas. To effectively distinguish them, we propose a lightweight essential/non-essential area detector (ENED) (Fig. 1 *Bottom*). The losses of non-essential features do not impact the model's predictions, whereas removing essential features causes the predictions unreliable. This difference allow us to determine whether the attacked area is essential or non-essential by observing the model's output . Specifically, after the attack detector, we obtain the mask **M** of adversarial patches, with which we can use $k$ neighbors of the adversarial patches to reconstruct $k$ images. After that, we input the reconstructed images into the model to obtain the corresponding output probabilities. At last, we compute the average similarity between each pair of outputs (Eq. 4),

$$\overline{similarity_{out}} = \frac{\sum_{i,j}^{k} sim(exp(out_i), exp(out_j))}{\binom{k}{2}} \qquad i,j \quad = 1\cdots\binom{k}{2}, i < j \qquad (4)$$

where $out_i$, $out_j$ refers to the output of the model for two different reconstructed images. When the average similarity $\overline{similarity_{out}}$ is below the predefined threshold $\gamma$, we deem that the attacked area is an essential area; otherwise, the attack occurs in the non-essential area. The ENED detection algorithm is shown in Appendix A.4.

### 3.3 ADAPTIVE DEFENSE FOR DIFFERENT ATTACK CATEGORIES

After we categorize adversarial patch attacks, we introduce corresponding defense methods for different types of attacks. We introduce these two different strategies below.

**Defense for non-catastrophic attacks (Reconstruction).** As mentioned above, removing adversarial patches is a superior defense for attacks occurred in non-essential areas. Specifically, utilizing the mask **M** (locating the adversarial patches), we can calculate the mask of neighbor patches (locating neighbor patches). After that, we randomly select a neighbor patch and use it to fill the masked adversarial patches, through which we get a reconstructed image. Eventually, we feed the reconstructed image into the original ViT architecture to obtain the final prediction.

**Defense for catastrophic attacks (TAS).** As mentioned in the Section 3.2, attention suppression method, like RTA (Yu et al., 2023), is an effective approach for attacks occurred in essential areas. Specifically, RTA restricts the attention of each token with unified global attention (i.e., mean of all token attention weights) to prevent the model from being misled by adversarial patches. However, calculating the mean of all token attentions is inevitably affected by the adversarial patches, making it difficult to effectively suppress the attention weights of adversarial tokens. Moreover, since some research (Vaswani, 2017; Han et al., 2022) has demonstrated that different attention heads stand for different representation subspaces, using a unified global attention for all heads is suboptimal for attention suppression. To solve the first problem, we remove the adversarial tokens when calculating

Table 2: **Comparison of different defense methods.** We report the model's average clean accuracy and robust accuracy (%) across various sizes of attack patches. The results demonstrate that our method has achieved exceptional robust performance under a range of attacks. The best results for each attack are marked in bold, and the second best are underlined.

| #Params. | | <20M | | <50M | | | | | <100M | | | | >100M |
|---|---|---|---|---|---|---|---|---|---|---|---|---|---|
| Model | | DeiT-T | BiF-T | DeiT-S | ViT-S | BiF-S | TNX-T | TNX-S | DeiT-B | ViT-B | BiF-B | TNX-B | ViT-L |
| | No defense | 71.1 | 81.6 | 78.8 | 74.2 | 83.5 | 83.8 | 84.5 | 81.1 | 80.3 | 84.0 | 84.7 | 84.9 |
| | RTA | 65.8 | 69.4 | 75.4 | 72.1 | 69.8 | 79.4 | 81.6 | 79.3 | 76.4 | 74.3 | 81.3 | 79.5 |
| No attack | Jedi | 71.8 | 80.6 | 78.4 | 73.2 | 82.4 | 82.6 | 83.8 | 80.3 | 79.2 | 83.6 | 84.1 | 84.2 |
| | ARMRO | 64.0 | 71.4 | 75.3 | 65.4 | 72.5 | 79.5 | 80.5 | 78.9 | 74.5 | 83.1 | 81.3 | 80.5 |
| | **NeighborViT** | **71.9** | **81.6** | **78.8** | **74.1** | **83.5** | **83.6** | **84.4** | **81.1** | **80.2** | **84.0** | **84.6** | **84.8** |
| | No defense | 0.1 | 5.3 | 0.0 | 0.0 | 6.3 | 10.2 | 10.4 | 0.4 | 0.6 | 6.4 | 11.1 | 1.3 |
| | RTA | 52.8 | 64.2 | 52.9 | 43.1 | 56.5 | 72.4 | 71.4 | 55.8 | 56.3 | 59.6 | 72.4 | 66.3 |
| ViTRPP | Jedi | 69.1 | 77.8 | 75.6 | 70.1 | 81.5 | 68.3 | 69.0 | 77.2 | 74.7 | 79.7 | 73.1 | 83.1 |
| (Gu et al., 2022) | ARMRO | 66.8 | 68.1 | 74.7 | 64.5 | 69.0 | 73.5 | 75.9 | 72.8 | 74.0 | 70.1 | 76.8 | 78.8 |
| | **NeighborViT** | **70.1** | **79.6** | **77.3** | **71.1** | **82.4** | **78.2** | **82.6** | **79.0** | **75.9** | **81.1** | **81.7** | **83.8** |
| | No defense | 0.5 | 7.1 | 1.4 | 0.0 | 9.7 | 12.4 | 15.8 | 6.0 | 3.1 | 9.4 | 13.6 | 4.9 |
| | RTA | 51.1 | 63.2 | 53.8 | 41.3 | 61.9 | 69.9 | 68.1 | 55.7 | 54.3 | 65.3 | 70.9 | 64.6 |
| Patch-F | Jedi | 66.2 | 64.3 | 66.4 | 70.8 | 65.8 | 67.0 | 76.2 | 64.8 | 71.1 | 71.2 | 73.0 | 76.9 |
| (Fu et al., 2022) | ARMRO | 64.7 | 67.2 | 72.9 | 67.3 | 69.7 | 71.3 | 74.9 | 68.5 | 70.5 | 72.8 | 78.3 | 77.1 |
| | **NeighborViT** | **67.9** | **73.3** | **74.6** | **72.2** | **78.4** | **79.1** | **79.8** | **75.1** | **73.1** | **79.2** | **81.3** | **80.9** |
| | No defense | 0.0 | 6.8 | 0.3 | 0.0 | 8.9 | 11.7 | 13.6 | 4.3 | 2.2 | 9.1 | 12.4 | 3.7 |
| | RTA | 49.1 | 61.8 | 50.6 | 40.6 | 59.6 | 67.8 | 67.2 | 52.3 | 51.4 | 63.2 | 68.5 | 61.2 |
| Attention-F | Jedi | 65.3 | 65.6 | 67.0 | 70.1 | 61.6 | 72.8 | 75.6 | 69.1 | 75.8 | 69.2 | 74.1 | 77.8 |
| (Lovisotto et al., 2022) | ARMRO | 66.4 | 69.1 | 74.2 | 68.6 | 66.2 | 72.5 | 74.5 | 71.4 | 73.6 | 71.0 | 79.8 | 78.4 |
| | **NeighborViT** | **70.3** | **77.6** | **77.4** | **71.8** | **79.5** | **80.8** | **81.3** | **79.1** | **76.6** | **80.9** | **81.2** | **81.7** |

the mean values of attention weights to suppress the attention weights of adversarial tokens more effectively. To deal with the second problem, we compute the mean values of attention weights separately for each head. The resulting attention suppression method is illustrated by Eq. 5,

$$TAS(\mathbf{A}_{i,j,h}^{(m)}) = \frac{\mathbf{A}_{i,j,h}^{(m)}}{\mathbf{A}_{j,h}^{(m)}} min(\mathbf{A}_{j,h}^{(m)}, \alpha \overline{\mathbf{A}_{mask_h}^{(m)}})$$

$$\overline{\mathbf{A}_{mask_h}^{(m)}} = \frac{1}{N^2 - win_{adv}^2}\sum_{i,j}\mathbf{A}_{i,j,h}^{(m)}, (i,j) \neq p_{adv}$$

(5)

where we denote $\mathbf{A}^{(m)}$ as the attention weight of the $m_{th}$ block of the ViTs. The $i, j$ represent the index of attention values and $h$ is the index of different heads. The $\mathbf{A}_{j,h}^{(m)} = \frac{1}{N}\sum i\mathbf{A}_{i,j,h}^{(m)}$ represents the average attention contribution of the $j_{th}$ token and $\alpha$ represents the suppression coefficient. We denote the average attention weight of clean tokens as $\overline{\mathbf{A}_{mask_h}^{(m)}}$, where $N$ denotes the total number of patches. The $win_{adv}$ is the attack patch size and $p_{adv}$ is the index of adversarial patches detected by the attack detector (can be found in Appendix A.4).

## 4 EXPERIMENTAL EVALUATION

### 4.1 SETUP

**Models and Datasets.** In our experiments, the target models we select are from the ViT family (ViT-S, ViT-B, and ViT-L) (Dosovitskiy et al., 2021), the DeiT family (DeiT-T, DeiT-S, and DeiT-B) (Touvron et al., 2021), the BiFormer family (BiFormer-T, BiFormer-S, and BiFormer-B) (Zhu et al., 2023), and the most recent TransNeXt family (TransNeXt-T, TransNeXt-S, and TransNeXt-B) (Shi, 2024). We utilize the official pre-trained versions of all selected models, employing an input patch size of $16 \times 16$ pixels. To generate adversarial patch attacks, we utilize the validation set of ImageNet (Deng et al., 2009) as the clean image dataset and then apply various attack strategies under different ViTs on these clean images. We set a wide range for the attack patch size, ranging from 0.5% to 8% of the total pixel area of the image. Specifically, we define $attack\ patch\ size = 1\times, 2\times, 3\times, 4\times$, representing that the side length of the attack patches is 1, 2, 3, and 4 times the input patch length (e.g., $2\times$ represents an attack patch size of $32 \times 32$ pixels).

**Attack Strategies.** To demonstrate the effectiveness of our method, we employ multiple state-of-the-art attack methods in our experiment, including *1) ViTRPP* (Gu et al., 2022), *2) Patch-Fool* (Fu et al., 2022), and *3) Attention-Fool* (Lovisotto et al., 2022). As all these methods are white-box attacks, they represent the most powerful ViT attack strategies currently available. The parameter

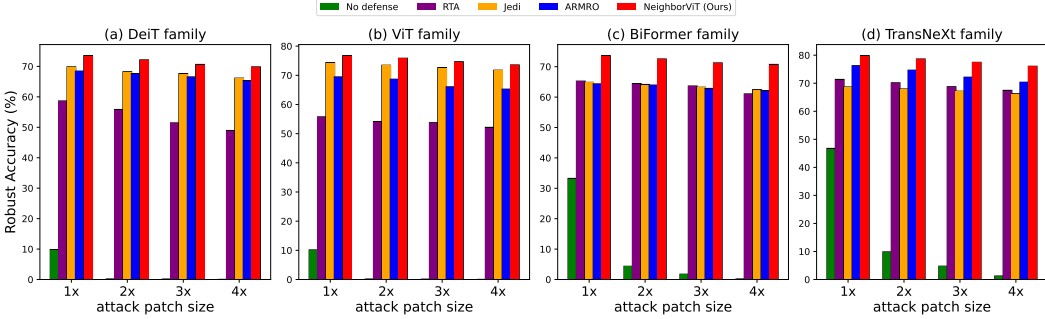

Figure 3: **Average robust accuracy (%) of different model families under various sizes of adversarial patches**. Each value is averaged across all models within each ViT family. $attack\ patch\ size = 1\times, 2\times, 3\times, 4\times$ denotes that the side length of the attack patch is 1, 2, 3, and 4 times the input patch length of the target model.

settings for all these attack strategies are identical to those in their original papers. The detailed attack settings are presented in Appendix A.5.

**Baseline Defense Methods.** In our experiments, we compare our method with several representative robust ViT frameworks, including RTA (Yu et al., 2023), ARMRO (Liu et al., 2023), and Jedi (Tarchoun et al., 2023), of which ARMRO and Jedi are the current state-of-the-art defense approaches. The detailed configurations of these baseline defense methods are presented in Appendix A.6.

**Evaluation Metrics.** To evaluate our proposed framework, we employ two metrics to assess its performance: *1) Clean Accuracy*: This metric measures the percentage of correctly classified images within the clean image dataset that has not been modified by any attacker. *2) Robust Accuracy*: This metric assesses the model's resilience to adversarial attacks. It measures the percentage of correctly classified images under adversarial patch attacks.

## 4.2 COMPARISON OF DIFFERENT DEFENSE METHODS

In this section, we comprehensively compare our defense method with RTA, Jedi, and ARMRO. We first randomly select 5,000 clean images from the validation set of Imagenet to evaluate the defended ViTs' clean accuracy. As shown in Table 2, our method achieves the highest clean accuracy, with an average reduction of less than 0.1% compared to the original model. This marks a significant improvement over Jedi, which has the smallest reduction among the previous methods, with a decrease of 0.8%. The superior performance of our framework is attributed to its differentiated handling of clean samples and adversarial patch attacks, and the high effectiveness of the attack detector.

Next, we assess the robust accuracy of the defended ViTs on adversarial images generated using various patch attack methods under multiple attack patch sizes. Table 2 presents different models' average robust accuracy across various attack patch sizes for each attack strategy. Meanwhile, Fig. 3 illustrates the average robust accuracy of different model families under various sizes of adversarial patches. As shown in Fig. 2 and Tab. 3, our method consistently outperforms others across all ViT models, attack methods, and attack patch size values, demonstrating its effectiveness in countering adversarial patch attacks. This success can be attributed to our adaptive defensive strategies against different types of adversarial attacks (catastrophic attacks and non-catastrophic attacks). Notably, our approach shows more significant improvement in defense against Patch-Fool and Attention-Fool compared to ViTRPP. This is because the adversarial patches generated by the former two strategies are more evenly distributed across essential and non-essential areas. When a specific type of attack dominates, the performance of our framework converges to that of a single defense method (e.g., when all attacks are catastrophic, the performance of our framework is comparable to that of RTA). In such cases, the improvement provided by our adaptive defense mechanism becomes limited. In real-world scenarios, since it is uncertain whether the input patch attack will be catastrophic or non-catastrophic, only our adaptive method can consistently achieve effective defense.

## 4.3 EVALUATION OF THE KEY COMPONENTS OF NEIGHBORVIT

In this section, we evaluate the effectiveness of the key components (attack detector, essential/non-essential area detector, and token attention suppression) in our defense framework.

| Model | Strategy | ViTRPP | | Patch-F | | Attention-F | |
|---|---|---|---|---|---|---|---|
| | | 2× | 4× | 2× | 4× | 2× | 4× |
| DeiT-S | TAS Only | 55.6 | 52.4 | 54.7 | 52.9 | 56.7 | 53.2 |
| | NR Only | 69.4 | 68.1 | 66.7 | 64.2 | 67.4 | 64.3 |
| | **Both** | **77.7** | **76.5** | **75.7** | **72.1** | **76.5** | **71.8** |
| ViT-S | TAS Only | 48.6 | 45.3 | 46.7 | 45.8 | 51.3 | 49.7 |
| | NR Only | 64.2 | 62.7 | 65.6 | 62.9 | 67.4 | 65.7 |
| | **Both** | **71.2** | **70.0** | **73.3** | **71.6** | **72.8** | **70.1** |
| BiF-S | TAS Only | 58.7 | 56.9 | 64.2 | 62.3 | 66.9 | 63.7 |
| | NR Only | 74.3 | 75.2 | 67.0 | 68.4 | 75.2 | 76.3 |
| | **Both** | **81.7** | **82.4** | **78.3** | **76.6** | **78.9** | **77.1** |
| TNX-S | TAS Only | 72.4 | 72.9 | 71.8 | 69.7 | 72.6 | 69.4 |
| | NR Only | 77.4 | 78.8 | 74.2 | 74.6 | 76.2 | 75.4 |
| | **Both** | **82.7** | **82.6** | **79.2** | **77.8** | **81.9** | **79.6** |

| Model | Strategy | ViTRPP | | Patch-F | | Attention-F | |
|---|---|---|---|---|---|---|---|
| | | 2× | 4× | 2× | 4× | 2× | 4× |
| DeiT-S | No defense | 49.4 | 46.2 | 50.3 | 47.7 | 51.8 | 46.2 |
| | RTA | 69.6 | 68.7 | 67.2 | 65.3 | 69.4 | 66.7 |
| | **TAS** | **77.7** | **76.5** | **75.7** | **72.1** | **76.5** | **71.8** |
| ViT-S | No defense | 53.7 | 47.4 | 52.5 | 50.2 | 54.8 | 52.7 |
| | RTA | 66.7 | 65.3 | 66.8 | 64.7 | 69.1 | 67.7 |
| | **TAS** | **71.2** | **70.0** | **73.3** | **71.6** | **72.8** | **70.1** |
| BiF-S | No defense | 62.1 | 59.7 | 57.6 | 55.8 | 63.7 | 62.4 |
| | RTA | 76.2 | 74.9 | 68.2 | 67.1 | 77.1 | 75.0 |
| | **TAS** | **81.7** | **82.4** | **78.3** | **76.6** | **78.9** | **77.1** |
| TNX-S | No defense | 65.3 | 62.7 | 63.1 | 61.8 | 67.4 | 65.3 |
| | RTA | 78.5 | 78.9 | 75.3 | 74.9 | 78.2 | 77.3 |
| | **TAS** | **82.7** | **82.6** | **79.2** | **77.8** | **81.9** | **79.6** |

Table 3: Ablation study of the essential/non-essential area detector. The best results are marked in bold.

Table 4: Comparison of different attention suppression methods on catastrophic attacks. The best results are marked in bold.

**Effectiveness of the Attack Detector.** The attack detection strategies used in current ViTs defense methods either need to train an auxiliary model (Tarchoun et al., 2023) or query the target model multiple times (Xiang et al., 2022; Yang et al., 2023), both of which are computationally expensive. In contrast, our attack detector is more accurate and lightweight. To demonstrate this, we compare the detection methods proposed in PatchCleanser (Xiang et al., 2022), IBCD (Yang et al., 2023), and Jedi (Tarchoun et al., 2023) with ours, evaluating both detection accuracy and average time cost for each sample. For the test dataset, we select a mixture of 500 clean and 500 adversarial patch attacks generated with Attention-Fool

Figure 4: Comparison of different attack detection methods. Our attack detector has the highest detection accuracy with little time cost.

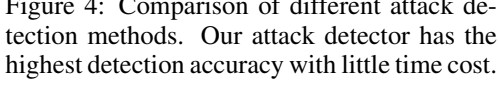

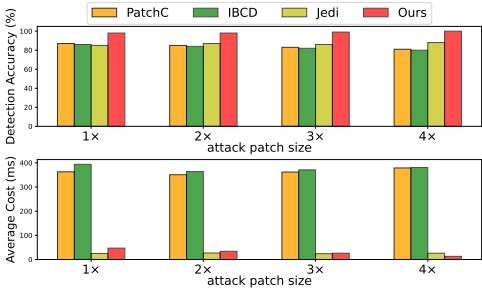

toward DeiT-X. As shown in Fig. 4, our attack detector achieves the highest detection accuracy across various attack patch sizes (with improvements of 10% to 12%) while keeping the time cost among the lowest. Additionally, as the attack patch size decreases, the time cost increases due to the search process starting from larger patches and progressively narrowing to smaller ones.

**Necessity and Efficacy of the Essential/Non-Essential Area Detector.** To demonstrate the necessity and efficacy of our essential/non-essential area detector (ENED), we conduct experiments comparing three defense strategies: *1) TAS Only*, where adversarial examples bypass ENED and are handled solely by the token attention suppression method; *2) NR Only*, where adversarial examples bypass ENED and are processed using the neighbor replacement method; and *3) Both*, where ENED categorizes adversarial patch attacks into catastrophic attacks and non-catastrophic attacks, subsequently handling them with TAS and NR, respectively. For the test dataset, we select one model from each ViT family and perform different attack strategies with two attack patch sizes. We then apply the three defense strategies to the target models and evaluate the robust accuracy of the models. As shown in Table 3, models with activated ENED achieve the highest robust performance, validating the efficiency and necessity of ENED.

**Effectiveness of Token Attention Suppression.** To enhance the robustness of ViT against catastrophic attacks, we propose token attention suppression (TAS), which applies fine-grained token attention suppression using masked global token information (attention of non-attacked patches). In this section, we assess the impact of different attention suppression strategies' influence on the robust accuracy of the defended models. We compare three approaches: no attention processing, RTA's restriction method, and our TAS, where we keep the non-catastrophic defense approach the same. As shown in Table 4, TAS achieves the best robust performance, with nearly a 7% improvement on DeiT-S and approximately a 5% improvement on ViT-S, BiF-S, and TNX-S, highlighting its effectiveness against catastrophic attacks.

## 4.4 HYPER-PARAMETERS

To evaluate the robustness of our proposed NeighborViT architecture, we conduct comprehensive studies by varying the hyper-parameters within our framework and compare the defensive effectiveness with the state-of-the-art method, Jedi. We employ the DeiT-T model architecture as the target model and randomly select 2,500 clean images to generate adversarial patch attacks for validation.

**Influence of $\varepsilon$.** We evaluate the detection threshold $\varepsilon$ in our attack detector, which determines whether an input sample is classified as an adversarial example. For this experiment, we utilize multiple attack strategies with $attack\ patch\ size = 2$. We adjust $\varepsilon$ and assess both the clean accuracy and robust accuracy of the defended model. As shown in Table 5, when $\varepsilon$ is less than 2.25, our method consistently outperforms Jedi in robust accuracy across all attack scenarios. Additionally, for $\varepsilon$

Table 5: The impact of different AD thresholds $\varepsilon$ on clean accuracy and robust accuracy. We report CA and RA under different attack methods. CA: clean accuracy; RA: robust accuracy. The results of our defense method, which are better than those of Jedi, are marked in bold.

| $\varepsilon$ | | 1.95 | 2.00 | 2.10 | 2.15 | 2.25 | 2.35 | 2.45 | 2.55 | Jedi |
|---|---|---|---|---|---|---|---|---|---|---|
| ViTRPP | CA | 67.2 | 68.4 | **71.6** | **72.4** | **72.4** | **72.4** | **72.4** | **72.4** | 69.8 |
| | RA | **70.2** | **70.3** | **71.1** | **71.6** | **68.6** | 66.3 | 64.7 | 62.8 | 68.4 |
| Patch-F | CA | 68.4 | 69.3 | **69.9** | **71.7** | **72.4** | **72.4** | **72.4** | **72.4** | 69.7 |
| | RA | **71.7** | **71.9** | **71.6** | **71.6** | **71.6** | **68.2** | 64.5 | 61.3 | 65.5 |
| Attention-F | CA | 68.9 | **70.3** | **72.4** | **72.4** | **72.4** | **72.4** | **72.4** | **72.4** | 70.0 |
| | RA | **71.6** | **71.7** | **71.6** | **67.1** | **65.3** | 62.6 | 60.7 | 59.8 | 64.7 |

greater than 2.10, it maintains a clear advantage over Jedi in clean accuracy. When $\varepsilon$ is between 2.10 and 2.25, our method consistently outperforms Jedi in terms of both clean accuracy and robust accuracy across the three attack scenarios.

**Influence of $\gamma$.** We assess the detection threshold $\gamma$ of our essential/non-essential area detector for categorizing adversarial inputs. For this experiment, we utilize Patach-Fool as the attack strategy with multiple attack patch sizes. We adjust $\gamma$ and evaluate the robust accuracy of the defended model. The results are shown in Table 6. Under each

Table 6: The impact of different ENED thresholds $\gamma$. The results better than Jedi are marked in bold. $1\times, \ ... \ , 4\times$ denotes $attack\ patch\ size = 1\times, \ ... \ , 4\times$.

| $\gamma$ | | 2.0 | 2.15 | 2.25 | 2.35 | 2.45 | 2.55 | 2.65 | Jedi |
|---|---|---|---|---|---|---|---|---|---|
| DeiT-T | $1\times$ | 61.4 | 62.9 | 64.7 | **68.5** | **70.3** | **71.6** | **69.3** | 66.0 |
| | $2\times$ | 60.8 | 63.3 | **66.2** | **69.8** | **71.6** | **68.1** | 64.3 | 65.5 |
| | $3\times$ | **65.4** | **68.2** | **70.5** | **71.6** | **67.9** | **66.1** | 62.7 | 64.3 |
| | $4\times$ | **64.8** | **71.6** | **67.9** | **67.4** | **64.2** | 61.8 | 59.7 | 62.5 |

attack patch size, our approach outperforms Jedi within a $\gamma$ range that spans over 0.3. When $\gamma$ is between 2.35 and 2.45, our method consistently outperforms Jedi across all attack patch sizes.

**Influence of $\alpha$.** We aim to suppress abnormally high attention weights for patch attacks located in the essential area. To achieve this, we introduce a key attention coefficient parameter, $\alpha$, representing the scaling factor for the mean of the masked attention (i.e., without adversarial tokens). In this section, we aim to assess the impact of $\alpha$. For this experiment, we utlize

Table 7: The impact of different attention coefficient parameters $\alpha$. The results of our defense method, which are better than those of Jedi, are marked in bold.

| $\alpha$ | 0.85 | 0.95 | 1.00 | 1.05 | 1.15 | 1.20 | 1.25 | 1.30 | Jedi |
|---|---|---|---|---|---|---|---|---|---|
| DeiT-T | 63.7 | 64.2 | **66.3** | **69.7** | **71.6** | **70.5** | **68.7** | **66.9** | 65.5 |

Patach-Fool as the attack strategy with $attack\ patch\ size = 2$. We adjust $\alpha$ and assess the robust accuracy of the defended model, with the results shown in Tab. 7. Our approach achieves stronger defense performance than the best baseline Jedi across a wide $\alpha$ range (from 1.00 to 1.30).

## 5 CONCLUSION

In this work, we introduce NeighborViT, a novel defense framework for Vision Transformers (ViTs) designed to counter adversarial patch attacks. Unlike traditional defense methods that treat all input samples equally, NeighborViT categorizes different types of inputs and applies adaptive, tailored defense mechanisms. Specifically, NeighborViT employs an attack detector to identify potential attacks in input images and further classifies the detected adversarial examples into catastrophic or non-catastrophic attacks. The key to NeighborViT's ability to detect, categorize, and mitigate adversarial attacks lies in its strategic use of neighbor information at various stages. Experimental results on both classical and state-of-the-art ViTs demonstrate the effectiveness of our proposed method, achieving superior robust performance while maintaining clean accuracy.

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

# A APPENDIX

## A.1 ADVERSARIAL PATCH

Figure 5 shows some examples of adversarial patches.

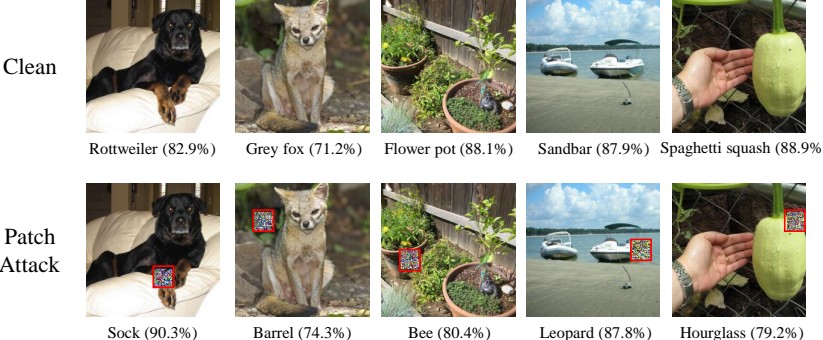

Figure 5: **Examples of patch attacks**. **First row:** Clean samples; **Second row:** Adversarial examples derived from various patch attack methodologies. Adversarial patches are highlighted with red boxes for better visualization.

## A.2 ADVERSARIAL PATCH DETECTION WITH SOBEL OPERATOR

Figure 5 shows some examples of adversarial patches detection with sobel operator.

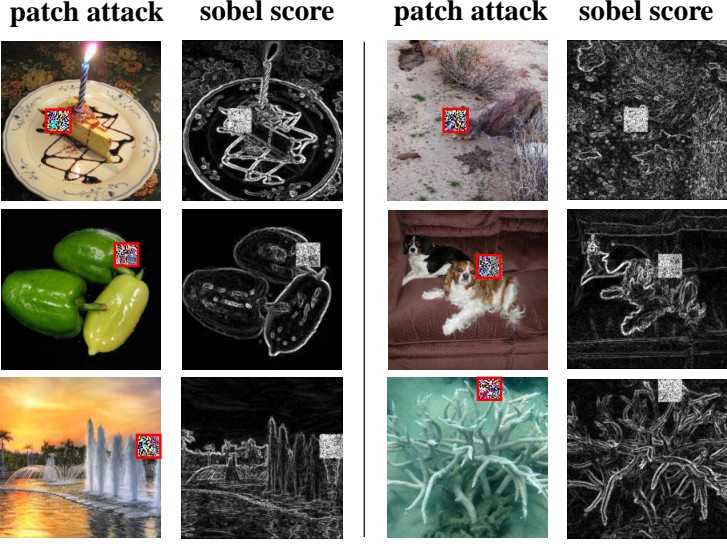

Figure 6: **The potential of sobel operator for adversarial patch attack detection**. We calculate the gradient of pixels on the image and white areas represent higher sobel scores.

## A.3 TAS & REMOVAL FOR CATASTROPHIC ATTACKS

In this section, we visualize the different effects of attention suppression (TAS) and removing adversarial patches on essential area attacks. We still analyze from the perspective of attention. Since the essential features contained in the essential area have been lost, our defense at this time should minimize the focus impact on other essential features. In Fig. 7, we visualize the changes in the attention of each layer for essential area attacks after using *1)* neighbor replacement (NR) to reconstruct the

image and *2)* using TAS for attention suppression. The selected samples are correctly classified with TAS but wrongly with NR.

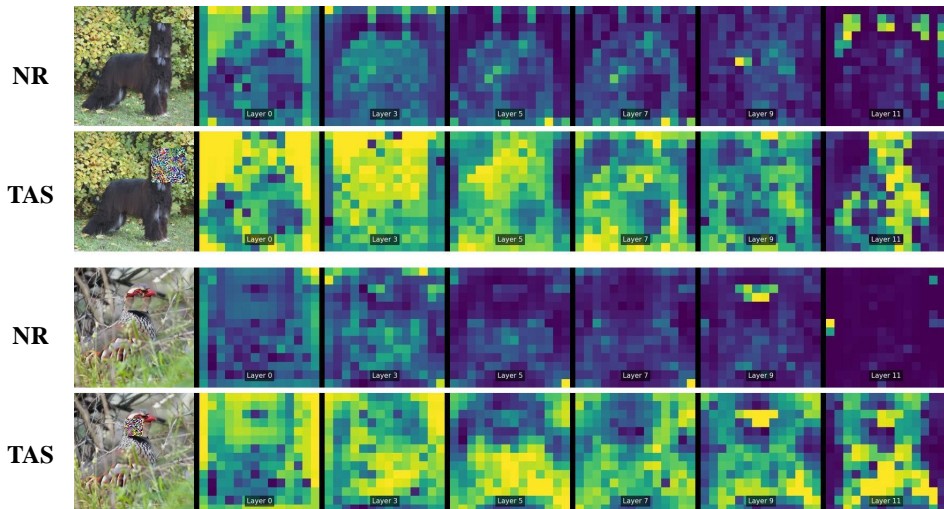

Figure 7: **Comparison between neighbor replacement construction (NR) and attention suppression with global attention algorithm (TAS).** The generation of the adversarial patches mainly changes their key vectors while the changes to the query and value are relatively small. The attention suppression method only suppresses the key vector and affects the attention calculation less; however, directly replacing the original adversarial patches with a neighbor will incur more significant effect on query and value vectors and affect the attention calculation more.

## A.4 ALGORITHMS

This section details the algorithmic principles of the attack detector (AD), essential/non-essential area detector (ENED). We have uniformly adopted the neighbor-informed mechanism in the design of these components. What differs is the type of neighbor information we consider in each component. In AD, We summarize our attack detection algorithm in Algorithm 1. For any given input $x$, we set the initial sliding window size to $\text{win}_h = \text{win}_w = 4$, beginning the detection from the top-left region of the image with a window stride equivalent to the model's patch size. After each window slide, we calculate the average sobel score of the patch within the current window. Concurrently, we sample neighbors in the four directions—top, bottom, left, and right—with the same size as the current window and compute the average score of all neighboring patches. After that, we calculate the distance of these scores from the current window's patch score. If the current distance exceeds the maximum distance $d_{max}$ updated in the previous instance, we update this calculation as the new maximum distance. Once a round of searching is completed, we obtain the maximum distance for that window size. If this distance surpasses a preset maximum distance $\varepsilon$, we consider the adversarial patches to have been detected and localized effectively and we can obtain the mask **M** of adversarial patches. Otherwise, we reduce the window size and proceed to the next round of searching. Since our distance measurements are patch-wise, we can design a uniform threshold $\varepsilon$ without dynamic variation for each attack methods.

In essential/non-essential area detection, we refer to the pixels of neighbor regions within the image. The implementation details of the algorithms are presented in Algorithm 2.

## A.5 ATTACK CONFIGURATIONS

We show the attack parameters in Tab. 8. We employ three attack scenarios: ViTRPP, Patch-Fool, and Attention-Fool. In each attack scenario, we set the perturbation area size with *attack patch size* $= 1\times, 2\times, 3\times, 4\times$. T-iters represents the total number of iterations. $lr$ represents the learning rate for the generation of adversarial patches. $\#steps$ and gamma denotes that for every $\#steps$ epoch, the learning rate is multiplied by gamma (which is typically less than 1) to

---

**Algorithm 1** Attack Detection with Sobel Operator

---

**Input:** input $x$; $win\_size$: the size of the sliding window; SOB: sobel operator; N: total number of patches; $\varepsilon$: preset distance threshold;

**Output:**

    $attack\_flag$          $\triangleright$ 0 No attack; 1 Adversarial examples;

    $p_{adv}, win_{adv}, \mathbf{M}$     $\triangleright$ $p_{adv}$: index of adversarial patches; $win_{adv}$: attack patch size; $\mathbf{M}$: mask of adv. patches.

1: initial:

2:     $s_{cur} \leftarrow 0$;          $\triangleright$ current window's average score

3:     $s_{nei} \leftarrow 0$;          $\triangleright$ neighboring patches' average score

4:     $d \leftarrow 0$;          $\triangleright$ distance of the current window and neighboring patches

5:     $d_{max} \leftarrow 0$;          $\triangleright$ maximum distance

6:     $win_{adv} \leftarrow 0$;          $\triangleright$ the attack patch size

7: sobel detection: $S_{(x)} \leftarrow SOB(x)$;

8: **while** $win\_size > 0$ **do**

9:     initial $d_{max} \leftarrow 0$;

10:     **for** $i \leftarrow 0; i < (\sqrt{N} - win\_size + 1)^2; i++$ **do**

11:         $s_{cur\_i} \leftarrow \frac{\sum s}{win\_size^2}$;          $\triangleright$ current window patches' sobel

12:         $s_{nei\_i} \leftarrow \frac{\sum neighbor}{4*win\_size^2}$;          $\triangleright$ neighboring patches' score

13:         $d \leftarrow s_{cur\_i} - s_{nei\_i}$;

14:         **if** $d > d_{max}$ **then**

15:             $p_{adv} \leftarrow i, d_{max} \leftarrow d, win_{adv} \leftarrow win\_size$;

16:         **else**

17:             slide to the next window;

18:         **end if**

19:     **end for**          $\triangleright$ If the threshold is exceeded, an attack is detected

20:     **if** $d_{max} > \varepsilon$ **then return** $attack\_flag = 1$ (mask $\mathbf{M}$, $p_{adv}, win_{adv}$)

21:     **else**

22:         $win\_size \leftarrow win\_size - 1$;

23:     **end if**

24: **end while**

25: **return** $attack\_flag = 0$;

---

**Algorithm 2** ENED Algorithm

---

**Input:** Input $x$, Adv mask $\mathbf{M}$, Neighbor mask $\mathbf{M}_n$, $f$ : the ViT model, $\gamma$: preset similarity threshold;

**Output:** attack in essential area (EA) or attack in non-essential area (NEA)

1: Get $k$ Neighbor mask: $\mathbf{M}_{n_1}, \mathbf{M}_{n_2}, \ldots, \mathbf{M}_{n_k}$

2: Get $k$ reconstructed image: $x_i' = \mathbf{M} \oplus x \odot \mathbf{M}_{n_i}$

3: Set $Sim\_sum \leftarrow 0$;

4: **for** $i \leftarrow 0; i < k-1; i++$ **do**

5:     **for** $j \leftarrow i+1; i < k; j++$ **do**

6:         $sim\_sum = sim\_sum + cos\_sim(f(x_i'), f(x_j'))$

7:     **end for**

8: **end for**

9: $\overline{sim} = \frac{sim\_sum}{\binom{k}{2}}$

10: **if** $\overline{sim} > \gamma$ **then**

11:     **Return NEA**

12: **else**

13:     **Return EA**

14: **end if**

---

reduce it progressively. In Patch-Fool, $\alpha$ is the coefficient for the attention loss, and #l represents the selection of the attention layer from which to optimize the adversarial patch. In Attention-Fool, $\alpha$ is the step size of PGD.

Table 8: Attack Parameters

| | Configurations |
|---|---|
| ViTRPP | cl;T-iters=500;lr=0.1;#steps=10;gamma=0.9 |
| Patch-Fool | cl;T-iters=250;$\alpha$=0.002;#l=4;lr=0.22;
#steps=10; gamma=0.95; |
| Attention-Fool | cl;T-iters=250;lr=0.25;$\alpha$=8/255;
#steps=10; gamma=0.95; |

## A.6 DEFENSE CONFIGURATIONS

In this section, we will present the detailed parameters of various baseline defense methods and our approach NeighborViT (Tab. 9, Tab. 10). In RTA, $\alpha$ is the restriction parameter. In JeDi, $\epsilon$ is the

Table 9: Baseline Defense Parameters

| | Configurations |
|---|---|
| RTA | $\alpha$=1.15 |
| JeDi | $\epsilon$=18.4;$r$=5 |
| ARMRO | $\tau$:
ViTs:1.43;
DeiTs:1.42;
BiFormers:1.57;
TransNeXts:1.62;
$cl$=1:Nd=1;
$cl$=2:Nd=5;
$cl$=3:2*(Nd=5);
$cl$=4:4*(Nd=5); |

entropy detection limit and $r$ neighbor sampling radius. In ARMRO, $\tau$ is the threshold to identify whether adversarial, and Nd is a preset coefficient stating the number of tokens needed to detect. In NeighborViT, $\varepsilon$ and $\gamma$ represent the attack detector (AD) and the essential/non-essential area detector (ENED) detection threshold, respectively. $\alpha$ represents the attention suppression coefficient parameters.

Table 10: NeighborViT Defense Parameters

| Model | $cl$=1 | $cl$=2 | $cl$=3 | $cl$=4 |
|---|---|---|---|---|
| ViTs | $\varepsilon$:
ViTRPP:2.15;
Patch-F:2.25;
Attention-F:2.10;
$\gamma$=1.93;
$\alpha$=1.05; | $\varepsilon$:
ViTRPP:2.15;
Patch-F:2.25;
Attention-F:2.10;
$\gamma$=1.89;
$\alpha$=1.05; | $\varepsilon$:
ViTRPP:2.15;
Patch-F:2.25;
Attention-F:2.10;
$\gamma$=1.82;
$\alpha$=1.05; | $\varepsilon$:
ViTRPP:2.15;
Patch-F:2.25;
Attention-F:2.10;
$\gamma$=1.75;
$\alpha$=1.05; |
| DeiTs | $\varepsilon$:
ViTRPP:2.15;
Patch-F:2.25;
Attention-F:2.10;
$\gamma$=2.55;
$\alpha$=1.15; | $\varepsilon$:
ViTRPP:2.15;
Patch-F:2.25;
Attention-F:2.10;
$\gamma$=2.45;
$\alpha$=1.15; | $\varepsilon$:
ViTRPP:2.15;
Patch-F:2.25;
Attention-F:2.10;
$\gamma$=2.35;
$\alpha$=1.15; | $\varepsilon$:
ViTRPP:2.15;
Patch-F:2.25;
Attention-F:2.10;
$\gamma$=2.15;
$\alpha$=1.15; |
| BiFormers | $\varepsilon$:
ViTRPP:2.15;
Patch-F:2.25;
Attention-F:2.10;
$\gamma$=2.33;
$\alpha$=1.27; | $\varepsilon$:
ViTRPP:2.15;
Patch-F:2.25;
Attention-F:2.10;
$\gamma$=2.25;
$\alpha$=1.27; | $\varepsilon$:
ViTRPP:2.15;
Patch-F:2.25;
Attention-F:2.10;
$\gamma$=2.18;
$\alpha$=1.27; | $\varepsilon$:
ViTRPP:2.15;
Patch-F:2.25;
Attention-F:2.10;
$\gamma$=1.97;
$\alpha$=1.27; |
| TransNeXts | $\varepsilon$:
ViTRPP:2.15;
Patch-F:2.25;
Attention-F:2.10;
$\gamma$=2.58;
$\beta$=5.21; | $\varepsilon$:
ViTRPP:2.15;
Patch-F:2.25;
Attention-F:2.10;
$\gamma$=2.24;
$\alpha$=1.32; | $\varepsilon$:
ViTRPP:2.15;
Patch-F:2.25;
Attention-F:2.10;
$\gamma$=2.13;
$\alpha$=1.32; | $\varepsilon$:
ViTRPP:2.15;
Patch-F:2.25;
Attention-F:2.10;
$\gamma$=2.07;
$\alpha$=1.32; |

