# OpenReview forum: "An Adaptive Defense Against Adversarial Patch Attacks For Vision Transformers"
_ICLR.cc/2025/Conference — ICLR 2025 Conference Withdrawn Submission_

### Official Review · Reviewer_aWJN · 2024-11-01

**Soundness:** 3
**Presentation:** 3
**Contribution:** 2
**Rating:** 3
**Confidence:** 3

**Summary:**

This paper investigates defense strategies against adversarial patch attacks aimed at Vision Transformers. The proposed method, NeighborViT, encompasses attack detection, classification of various attack types, and corresponding mitigation strategies. Specifically, the authors employ the sober operator with dynamic windows to pinpoint the locations of adversarial patches and use average predictions from the reconstruction of these patches to classify the type of attack. When adversarial patches are located in non-essential areas, they select a neighboring patch to fill the masked adversarial patches. In contrast, for essential areas, they reweight the attention weights for adversarial tokens. They conduct experiments on several ViTs to demonstrate the superior performance of their method.

**Strengths:**

1. The authors observe that different types of attacks require tailored defense approaches, classifying them as non-catastrophic and catastrophic based on the location of adversarial patches. This observation, validated by Table 1, demonstrates a logical approach that improves defense performance.
2. The authors conduct extensive experiments, which show better performance compared to existing defense.

**Weaknesses:**

1. The authors delve into adversarial defense within the filed of ViTs. the proposed defense methods present challenges when applied to defending CNNs. This limitation restricts the technological contribution and general application of the paper.

2. The proposed method includes three hyperparameters. While the authors present a wide range of values for these parameters, allowing the method to surpass other defenses. However,  there remains some limitation, as shown in Table 6. The optimal value of $gamma$ varies with different attack patch sizes.

These two weaknesses pose challenges to the practical application of the proposed adaptive method.

**Questions:**

The authors should consider adaptively tuning hyperparameters to enhance the method’s generalization.

Attack types are classified based on the location of adversarial patches, but could a more fine-grained classification further enhance detection and defense effectiveness?

---

### Official Review · Reviewer_Gych · 2024-11-01

**Soundness:** 2
**Presentation:** 3
**Contribution:** 2
**Rating:** 5
**Confidence:** 3

**Summary:**

The paper introduces NeighborViT, an adaptive defense framework for Vision Transformers (ViTs) against adversarial patch attacks. Existing defense methods sacrifice clean accuracy or achieve suboptimal robustness. NeighborViT detects and categorizes different types of attacks, applying tailored defense mechanisms. The framework leverages information from neighboring patches to enhance robust accuracy without compromising clean accuracy. Experimental results demonstrate its effectiveness on various ViT models using the ImageNet dataset.

**Strengths:**

This paper introduces an adaptive defense strategy that distinguishes between different types of adversarial attacks, providing tailored solutions, which successfully enhances robust accuracy while preserving clean accuracy, addressing a common trade-off in existing methods.
The authors provide extensive experimental results across multiple ViT models and attack approaches, demonstrating the framework's robustness.
The authors proposes a lightweight adversarial patch detection method that doesn’t require auxiliary models or multiple queries, reducing computational costs.

**Weaknesses:**

The experiments focus on specific attack types. Including a broader range of attacks could enhance the robustness of the evaluation.
Some technical aspects, such as the definition of similarity in Section 3.2, parameter cl in table 10, could be explained in greater detail.
The proposed strategy may require fine-tuning for different datasets or attack scenarios, potentially affecting generalizability.
The paper lacks discussions of the limitations of the proposed method.

**Questions:**

Since the proposed ENED relies on the detection results of AD, if naturalistic adversarial patches and other non-noise forms of adversarial patches are introduced for attacks on ViT later on, would the entire method become ineffective? In other words, does the low pixel continuity assumption hold true for these patches (e.g., TnT Attacks! Universal Naturalistic Adversarial Patches Against Deep Neural Network Systems) and (e.g., Generating transferable adversarial examples against vision transformers)?

In Table 1, the performance improvement of this method seems to be only around 2% compared to the secondary defense methods under most settings. Furthermore, Jedi can support multitasking (such as object detection) and various attacks (like naturalistic adversarial patches), suggesting that this method may have more scenario limitations.

Section 4.4 devotes a significant amount of space to discussing the values of hyperparameters. I am concerned that the impact of these hyperparameters may be too substantial. Further, do the hyperparameters need to be adjusted for different ViT architectures and datasets, and if so, how should this be approached?

The experiment does not specify the ratio of catastrophic attacks to non-catastrophic attacks, and it does not clearly differentiate which category each attack method belongs to.

For other suggestions or questions, please refer to weaknesses.

---

### Official Review · Reviewer_Vsut · 2024-11-01

**Soundness:** 3
**Presentation:** 3
**Contribution:** 2
**Rating:** 5
**Confidence:** 4

**Summary:**

This paper proposes a novel adaptive defense framework -- NeighborViT which can detects and categorizes different types of attacks; applies tailored defense mechanisms for each attack type and leverages information from neighboring patches for effective detection and defense.

**Strengths:**

1. The method analysis is clear and understandable.
2. The visualization is helpful.
3. The proposed method is simple but works well and maintain high clean accuracy with minimal reduction.

**Weaknesses:**

1. Lack of ablation study about different operators.
2. Lack of experiments about different attack strength.
3. Although the result shows the proposed method does not bring a lot extra computation cost, the detection and defense mechanisms can be further improved.
4. Bar graphs can be further improved (color, layout).

**Questions:**

1. It seems that using sobel operator is more like a emperical design, have you ever tried differen operators?
2. I am curious about the performance of the proposed method when attack patch with $L_p$ constraint.
3. Is that possible to make the dynamic scan of the patches and use more similar patch to replace but not just neighbour patches?

---

### Official Review · Reviewer_AZjc · 2024-11-02

**Soundness:** 2
**Presentation:** 2
**Contribution:** 2
**Rating:** 3
**Confidence:** 4

**Summary:**

This paper presents NeighborViT, a novel adaptive defense framework designed to counter adversarial patch attacks for ViTs. NeighborViT stands out by detecting and categorizing different types of attacks on inputs and applying adaptive, tailored defense mechanisms for each type of attack. Experimental results demonstrate that NeighborViT significantly enhances robust accuracy without compromising clean accuracy.

**Strengths:**

1. The framework of NeighborViT is detailed presented.
2. The idea of distinguishing different types of adversarial patch attacks and adopting corresponding defense methods is pretty interesting.

**Weaknesses:**

1. The attacks in experiments include one patch, such as Figure 5, while adopting corresponding defense methods for various attacks. It makes me quite confused. Therefore, I don't think the results can demonstrate the idea of adopting corresponding defense methods. For instance, experiments on adversarial examples with more adversarial patches should be performed to validate the performance of the proposed and baseline methods.
2. Since the attack detector and area detector are extra modules, the additional time cost (e.g., seconds per epoch), computational cost (e.g., FLOPs), and GPU memory usage of the framework should be carefully illustrated. In practice, training costs play an important part in the application. Improving adversarial robustness may not pay the overhead training cost without extra experiments. For example, authors can perform a detailed ablation study of each module and other baselines in their default settings.
3. The framework is an input process defense method, which aims at filtering out the adversarial perturbations for better adversarial robustness. In the manuscript, authors compare their method with some representative robust ViT frameworks. However, some classical input process defense methods are not considered in this paper, leading to thin arguments. For example, classical input process defense methods like smoothing, quantization, JPEG compression, and the recent work "Diffusion Models for Adversarial Purification" should be compared in their setting to figure out the effectiveness of their method.
4. The writing of the paper should be improved, which is unclear to me. For example, the illustration of "catastrophic" and "non-catastrophic" attacks is confusing. It is first proposed in the "Background & Related Works" without any explanation or reference. Only a simple introduction in L151-153 says "The catastrophic attacks represent the attacks occurred in the essential areas and non-catastrophic attacks represent the attacks located in the non-essential parts" with almost no information but two new words "essential" and "non-essential" parts. In subsection 3.2, the authors demonstrate that the "essential" parts contain essential features for model classification while the "non-essential" parts do not. However, there seems to be no detailed explanation of essential features, except from a plain description in L291-292. On the contrary, excessive words focus on how to get essential features by their area detector, leading to a reversed order. Papers can reveal their novelty through clever words but not rely on unintelligible sentences to cover their confused ideas.

**Questions:**

Please see the "Weakness" part for my questions.

---

### Note · Authors · 2024-11-15

I have read and agree with the venue's withdrawal policy on behalf of myself and my co-authors.